# Pervasive distribution of polyester fibres in the Arctic Ocean is driven by Atlantic inputs

Peter S. Ross [1,3✉], Stephen Chastain[1], Ekaterina Vassilenko[1], Anahita Etemadifar[1], Sarah Zimmermann[2], Sarah-Ann Quesnel[2], Jane Eert[2], Eric Solomon[1], Shreyas Patankar[1], Anna M. Posacka[1] & Bill Williams[2]

Microplastics are increasingly recognized as ubiquitous global contaminants, but questions linger regarding their source, transport and fate. We document the widespread distribution of microplastics in near-surface seawater from 71 stations across the European and North American Arctic - including the North Pole. We also characterize samples to a depth of 1,015 m in the Beaufort Sea. Particle abundance correlated with longitude, with almost three times more particles in the eastern Arctic compared to the west. Polyester comprised 73% of total synthetic fibres, with an east-to-west shift in infra-red signatures pointing to a potential weathering of fibres away from source. Here we suggest that relatively fresh polyester fibres are delivered to the eastern Arctic Ocean, via Atlantic Ocean inputs and/or atmospheric transport from the South. This raises further questions about the global reach of textile fibres in domestic wastewater, with our findings pointing to their widespread distribution in this remote region of the world.

[1] Ocean Wise Conservation Association, P.O. Box 3232, Vancouver, BC V6B 3X8, Canada. [2] Department of Fisheries and Oceans Canada, Institute of Ocean Sciences, P.O. Box 6000, Sidney, BC V8L 4B2, Canada. [3] Present address: Department of Earth, Ocean and Atmospheric Sciences, University of British Columbia, Vancouver, BC V6T 1Z4, Canada. ✉email: Peter.Ross@Ocean.org

Microplastics (MPs) have emerged as a significant global concern, having permeated the most remote reaches of the world[1–4]. MPs have been detected in Arctic pack ice[5], seawater[6–8] and seafloor sediments[9], but limited information exists on the mechanisms underlying their distribution and the scale of contamination. MPs may be expected to settle out over time onto sediments, where they will ultimately be buried[10]. Ingestion of MPs by numerous species have been documented across the various habitats around the world, from benthic to pelagic[9,11,12], and across all levels of the marine food web[13–15]. Laboratory-based studies suggest the potential for significant harm associated with MP ingestion, although the real-world health consequences for sealife remains unclear[16,17].

Early reports have suggested that processes including the North Atlantic Thermohaline Circulation[10], wave-driven Stokes drift[18], riverine input[6], and sea ice incorporation of MPs[5,19] contribute to the accumulation and movement of these emerging contaminants in the Arctic. The atmosphere remains a poorly understood mechanism for the transport of MPs, but past atmospheric pollution research points to this conduit as a potentially important pathway for MP transport into remote regions, including the Arctic[20–22]. While data are scant, reports of MPs in seafood have raised concerns about the potential for human ingestion and possible health effects[23]. For the indigenous peoples who rely heavily on foods from the ocean, including the Inuit inhabiting the circumpolar region, such concerns underscore the need for a more cohesive understanding of microplastic distribution and fate.

While MPs are defined as any particle of synthetic plastic smaller than 5 mm, the wide variety of shapes, sizes, colours and chemical composition among particles in the environment underlie a complex and diverse source function. And while primary MPs (e.g. microbeads and commercial pellets) are readily identifiable, the identities and sources of secondary MPs (i.e. those that have broken down from parent products) remain extraordinarily elusive. However, fibres represent a notable shape encountered in sediments and seawater samples[8,24].

Accurate identification of MP particles in the environment is complicated by the fact that plastics undergo time-dependent physico-chemical changes in the presence of UV radiation, oxygen, enzymes and digestive processes in biota[25–27]. These weathering changes to the surface of MPs can hamper their identification by Fourier transform infrared spectrometry (FTIR), but also have the potential to inform the characterization of MP transport and fate processes in the environment[25,28].

Differences in the minimum size dimensions of MPs captured by various studies is one fundamental constraint to comparing across studies. Simply put, small particles are difficult to sample and difficult to analyse. As a result, very few Arctic studies report on MPs at the low end of the size range ($<250 \, \mu m$), despite concerns about the potential for particles at this end of the size spectrum ($<20 \, \mu m$) to translocate tissues and bioaccumulate[29]. There is also evidence to suggest that smaller particles are also more abundant[30].

Here we characterize microplastic abundance, size and polymer identities throughout the waters of the Arctic Ocean, and at depths down to 1015 m at six sites in the Beaufort Sea. Our study provides foundational insights into the identity, transport and extent of MPs in the Arctic, and serves as a basis to characterize those source functions that can enable mitigation strategies. Our focus on sub-surface particles and their properties provide timely insights into the presence, movement and infra-red profiles of MPs in the Arctic environment, and sharpens the identification of future research needs.

## Results and discussion

We document here the presence, extent, shape and polymeric identities of MPs in seawater in the Arctic Ocean collected in 2016 during four oceanographic cruises at 71 stations and to depth in six vertical profiles ($n = 26$ samples) in the Beaufort Sea (Fig. 1). Stringent contamination protocols reduced or eliminated contamination of samples in the field and in the laboratory. The spatial assessment of MP distribution was derived from near-surface samples (3–8 m below the surface) to avoid a bias towards floating plastics and to enable an assessment of pelagic contamination across the Arctic Ocean. Micro-FTIR analysis of suspected MPs (37.6% or 590 of 1570 of SMPs were analysed by FTIR) enabled an adaptive approach to estimating total abundance within samples and across the Arctic, while eliminating those particles identified as (non-plastic) contaminants. The FTIR spectra also afforded us an opportunity to characterize the infra-red signatures of MPs across the Arctic environment and document broad patterns of microplastic distribution in the polar environment.

Counts of visually identified suspected microplastics (SMPs) in seawater averaged $186 \pm 15.4$ particles $m^{-3}$, the majority of which were fibres (Table 1). Confirmed MP counts derived by subsequent FTIR analysis of SMP particles led to a corrected Arctic-wide MP count of $40.5 \pm 4.4$ particles $m^{-3}$. The difference between suspected and confirmed MP counts underscores the vital importance of validating visual enumeration of MPs with subsequent spectroscopic analysis[31]. The size-frequency of MPs and SMPs revealed a skewed distribution which may partly reflect our choice of mesh size for sample collection (63 μm; Supplementary Fig. 2).

Our near-surface microplastic counts from sites across the Arctic fall into the range of those reported recently for Hudson Bay and the Eastern Arctic[4,7,8], but are higher than those measured in the polar waters of south and southeast region of Svalbard, Norway[6]. The dominance of fibres is consistent with the few studies that were able to verify the identity of synthetic fibres via FTIR imaging[6,7]. Of SMPs dominated by fibres, 41% were identified by FTIR as cellulosic. Since we had first applied visual criteria (i.e. via microscopic examination) to identify those particles that appeared synthetic, our subsequent FTIR-identification of cellulosic fibres leads us to conclude that some of these cellulosics originate from human-made materials, such as textiles. Together with previous reports of cellulosic fibres in the Arctic[2,8], there is a distinct need to characterize the abundance, distribution and fate of cellulosic materials in the environment, both natural and synthetic.

We observed a significant relationship between FTIR-confirmed MP concentration and longitude across the Arctic, with higher concentrations detected towards the East (Fig. 2A; $p < 0.001$; excluding the polar region $> 85°N$). The Atlantic-influenced, eastern Arctic (defined as sampling sites east of 105° W) had higher MP concentrations than the Pacific-influenced, western Arctic (defined as west of 105°W; Kruskal–Wallis $p < 0.001$; Dunn's test $p < 0.001$). The North Pole region (defined as above 85°N) did not differ from the eastern Arctic (Kruskal–Wallis $p < 0.001$; Dunn's test $p > 0.05$), but had higher MP counts than the western Arctic (Kruskal–Wallis $p < 0.001$; Dunn's test $p = 0.011$).

The length of MP fibres did not vary significantly with longitude ($R^2 = 0.221$, $p = 0.057$; Fig. 2B), but the length of SMPs did (Kruskal–Wallis $p < 0.001$; Dunn's $p < 0.001$). In addition, the length of MP fibres differed between East and West (Kruskal–Wallis $p = 0.045$; Dunn's $p < 0.001$), as did the length of SMPs (Kruskal–Wallis $p < 0.001$; Dunn's $p < 0.001$).

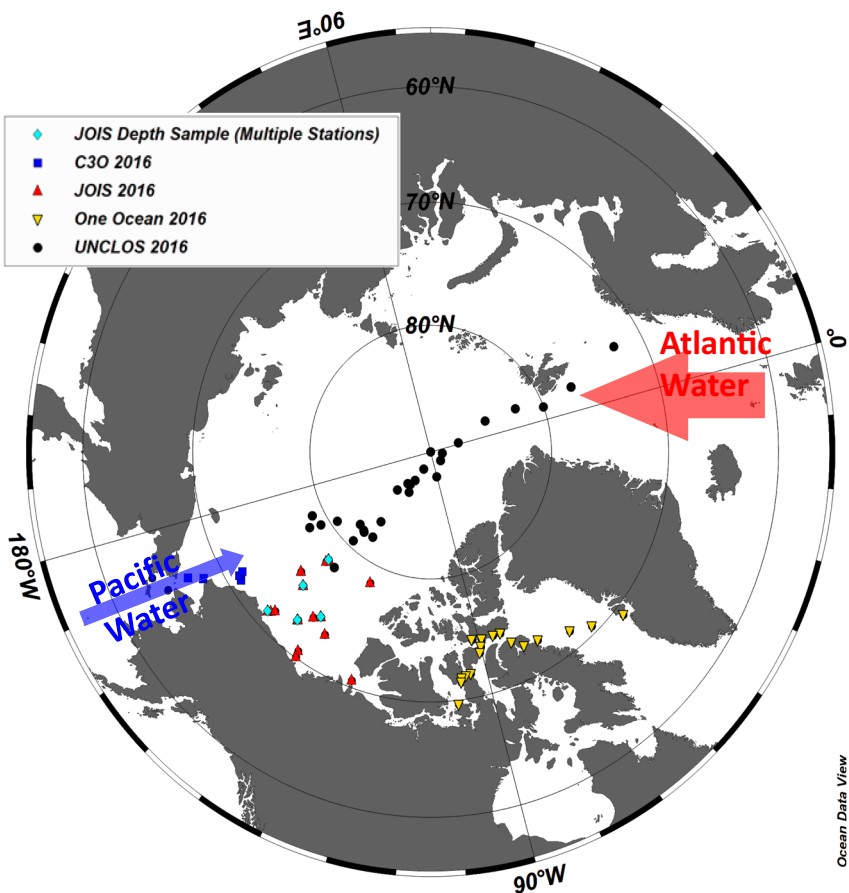

**Fig. 1 Microplastic samples were collected throughout the Arctic Ocean.** Microplastic (MP) particles were characterized in 71 near-surface (3–8 m depth) seawater samples collected during four oceanographic expeditions in 2016: (i) aboard the *CCGS Sir Wilfrid Laurier* with samples from the North Pacific Ocean, Bering Sea and Chukchi Sea (C30; blue squares); (ii) the United Nations Convention of the Law of the Sea expedition aboard the *CCGS Louis S. St-Laurent* along a transect from Tromsø, Norway, passing over the North Pole and into the northern Canada Basin (UNCLOS; black circles); (iii) the Joint Ocean Ice Study aboard the *CCGS Louis S. St Laurent* with samples from the Canada Basin (JOIS; red triangles); and (iv) the One Ocean Expeditions *RV Akademik Ioffe*, with samples collected from Greenland through the central Canadian Arctic Archipelago (OOE; yellow triangles). Microplastic samples (26) were collected at six stations in the Beaufort Sea down to 1015 m (pale blue diamonds). Arrows are drawn to provide an approximate representation of the well-described inflows of Atlantic- and Pacific-origin waters into the Arctic Ocean. The width of the arrows is proportional to the volume of the inflow: ~0.9 Sv Pacific water, ~8 Sv Atlantic water (1 Sv = $10^6$ m$^3$/s, e.g. Østerhus et al. [36]).

**Infra-red profiles of polyester fibres in the Arctic Ocean.** Fibres dominated MP particles across the Arctic (92.3%), with the majority of these being polyester (73.3%), having a relatively consistent width (14.1 μm) that resembled polyethylene terephthalate (PET) from textiles (Supplementary Table 1). While synthetic ropes and lines used in the fishing sector are a candidate source of concern[32], the dominant polymer types employed in fishing gear accounted for only 8.3% (nylon), 3.3% (polypropylene) and 0% (polyethylene) of MPs in our samples.

Differences in the infrared spectra provided us with an opportunity to evaluate factors affecting the fate of polyester fibres (see the "Methods" section). Time-dependent changes in spectral properties of materials due to weathering have been reported for several polymeric materials[25,33,34]. In a controlled laboratory-based year-long weathering study of commercial polyester textile samples, we observed gradual, time-dependent shifts in infrared spectra, which allowed us to develop and apply a quantitative metric to apply to distribution and fate of fibres and to inform on source and weathering functions (Supplementary Fig. 1).

The Peak Ratio Index derived here was calculated as the ratio of heights of a peak that varied over time and space (970 cm$^{-1}$) and a more stable reference peak (1240 cm$^{-1}$). We observed a significant relationship with longitude, with gradual changes towards the western Arctic (Fig. 2c; $R^2 = 0.235$; $p = 0.042$). In addition, average Peak Ratio Indices differed between the eastern and western Arctic regions (all regions Kruskal–Wallis $p = 0.044$; East-West Dunn's $p = 0.02$). The indices for polyester fibres in the eastern Arctic were in the range observed for unweathered (i.e. new) commercial polyester textile fibres we analysed at the laboratory, while fibres in the western Arctic exhibited lower indices, which we speculate to be related to a longer residence time in the environment. Additional research on weathering-related changes in ageing samples of polyester and other MPs will be a priority to inform aspects of microplastic source, transport and fate in the environment.

**MPs found throughout the water column in the Beaufort Sea.** Microplastic counts were found down to a depth of 1015 m at the six sites in the Beaufort Sea investigated here (Fig. 3). Suspected MP counts averaged 174 ± 21.2 m$^{-3}$ (range 26–427) throughout the water column, while FTIR-confirmed MP counts averaged 37.3 ± 6.9 m$^{-3}$ (range 0–200). A dominance of polyester was evident throughout the water column (71% of FTIR-confirmed MP, and 66% of plastic microfibres), highlighting the

**Table 1 Microplastic counts and properties in near-surface seawater samples across the Arctic Ocean.**

| Region | # Samples | Suspected microplastics[a] | | | | | FTIR-confirmed microplastics[b] | | | | | |
|---|---|---|---|---|---|---|---|---|---|---|---|---|
| | | Total SMPs counted | Concentration (SMP m⁻³) | % fibres | Fibre length (μm) | Fibre width (μm) | Total particles scanned (% of total) | Concentration (MP m⁻³) | % fibres | Fibre length (μm) | Fibre width (μm) | FTIR peak ratio index (PET)[c] |
| Eastern | 24 | 196 | 272 ± 22.1 (133–500) | 91.8% (71.4–100) | 1196 ± 69 (170–4250) | 14.5 ± 0.39 (2–40) | 97 (49.5) | 65.1 ± 8.0 (15.7–205) | 95.8% (0–100) | 1220 ± 162 (230–3375) | 14.8 ± 0.83 (8–40) | 0.28 ± 0.12 (0.001–2.22) |
| North Pole | 13 | 92 | 236 ± 29.3 (66.7–400) | 91.3% (81.8–100) | 827 ± 93 (135–4225) | 14.7 ± 0.68 (4–35) | 23 (25) | 44.3 ± 6.5 (9.9–107) | 100% | 1932 ± 622 (775–3630) | 11.3 ± 0.5 (10–12) | 0.47 ± 0.26 (0.111–4.13) |
| Western | 34 | 247 | 102.7 ± 17.0 (0–367) | 93.7% (66.7–100) | 897 ± 51.5 (45–4860) | 14.5 ± 0.46 (4–44) | 81 (32.8) | 21.1 ± 5.0 (0–153) | 93.8% (50–100) | 1044 ± 159 (230–3630) | 14.4 ± 0.8 (8–20) | 0.91 ± 0.26 (0.111–4.13) |
| Arctic-wide | 71 | 535 | 186 ± 15.4 (0–500) | 92.8% (66.7–100) | 1003 ± 39.2 (45–4860) | 14.9 ± 0.25 (2–60) | 201 37.6% | 49.0 ± 5.68 (0–260) | 91.4% (0–100) | 1132 ± 101 (230–3630) | 14.1 ± 0.6 (7–40) | 0.58 ± 0.13 (0.001–4.13) |

Data presented as mean ± SEM (range; min–max).
[a]Suspected microplastics (SMPs) were enumerated visually with light microscopy.
[b]Fourier transform infra red (FTIR)-confirmed MP counts represent adjusted counts following spectroscopy as described in the "Methods" section.
[c]The FTIR weathering index was calculated as the ratio between two peaks observed in polyester absorbance spectra at 970 and 1240 cm⁻¹, with low numbers observed for newer polymers.

pervasive spread of synthetic fibres throughout the waters of the Arctic Ocean.

MPs from depth-sampled stations in the Beaufort Gyre revealed water mass-related trends in their abundance. The concentration was higher in the Polar Mixed Layer (near-surface), lower around the core of the Pacific-origin water (200 m deep), and higher again around the core of the Atlantic-origin water (450–500 m deep). It is interesting to note that near-surface samples held a wider variety of polymer types, while deeper samples appear increasingly to be dominated by polyester. The surface waters in the Beaufort Gyre have been characterized as a mix of Pacific-origin water that is freshened by river runoff and ice melt. Interestingly, polyester was the dominant polymer in all water masses examined. In contrast to our findings, a recent meta-analysis of plastic polymers in the marine environment points to density-driven segregation of these particles within the water column[35]. Environmental weathering and biofouling may alter MP properties, highlighting the need to characterize weathering, porosity and microbial growth as additional metrics in MP research.

Our results coincide with the circulation of Atlantic, Pacific and fresh water in the Arctic Ocean. Approximately 8 Sv (1 Sv = $10^6$ m³/s) of salty Atlantic water flows north through Fram Strait and the Barents Sea[36] and then descends beneath fresher Arctic surface waters and circulates around the entire Arctic Basin[37–39]. Approximately 0.9 Sv of intermediate salinity Pacific water flows north through Bering Strait and across the Chukchi Sea shelf and then descends to around 50–200 m deep to lie beneath the fresher surface waters and above the salty Atlantic water[36,38,40,41]. Pacific water does not permeate the entire Arctic; it circulates around the Beaufort Gyre before exiting with the Atlantic water via the Canadian Arctic Archipelago and past Greenland[36,42,43].

While our MP profiles suggest a water mass association that is consistent with our near-surface MP counts and well-known Arctic Ocean circulation patterns, they are not conclusive. More sampling and 3-D numerical modelling would be required to distinguish between local inputs of MPs and those transported by oceanic flows or atmospheric transport. Local sinking of MPs, in combination with stratification may also give rise to variations with depth that we observed[44].

The North Pole region is near the edge of the influence of Pacific-origin water in the Arctic Ocean, and surface waters there have been found to be a mix of Pacific-origin and Atlantic-origin water and river water that has flowed from the Siberian shelves across the Arctic in the wind-forced and ice-forced trans-Arctic drift[42,45,46].

**Polyester fibres in the Arctic—from home laundry?** Home laundry is proving to be a potentially important conduit for the release of microfibres into aquatic environments[47–50]. We recently estimated that a single apparel item can release millions of fibres during a typical domestic wash[51]. The downstream implications are important; we also demonstrated that a single major secondary wastewater treatment plant can release as much as 21 billion microfibres into the receiving environment annually[52], with an estimated collective release of microfibres from all households in Canada and the USA of $3.5 \times 10^{15}$ microfibres (or 878 tonnes) annually[51]. These estimates follow reports of large numbers of microfibres being shed by various textiles in home laundry[49,50,53,54], and a dominance of synthetic microfibres in municipal wastewater[55]. Atlantic-origin water in the Arctic Ocean can be traced using isotopes released from Sellafield and La Hague, thus providing ample evidence that coastal inputs are mixed into the open ocean and contaminants readily dispersed[30,56,57]. While further inventories will no doubt

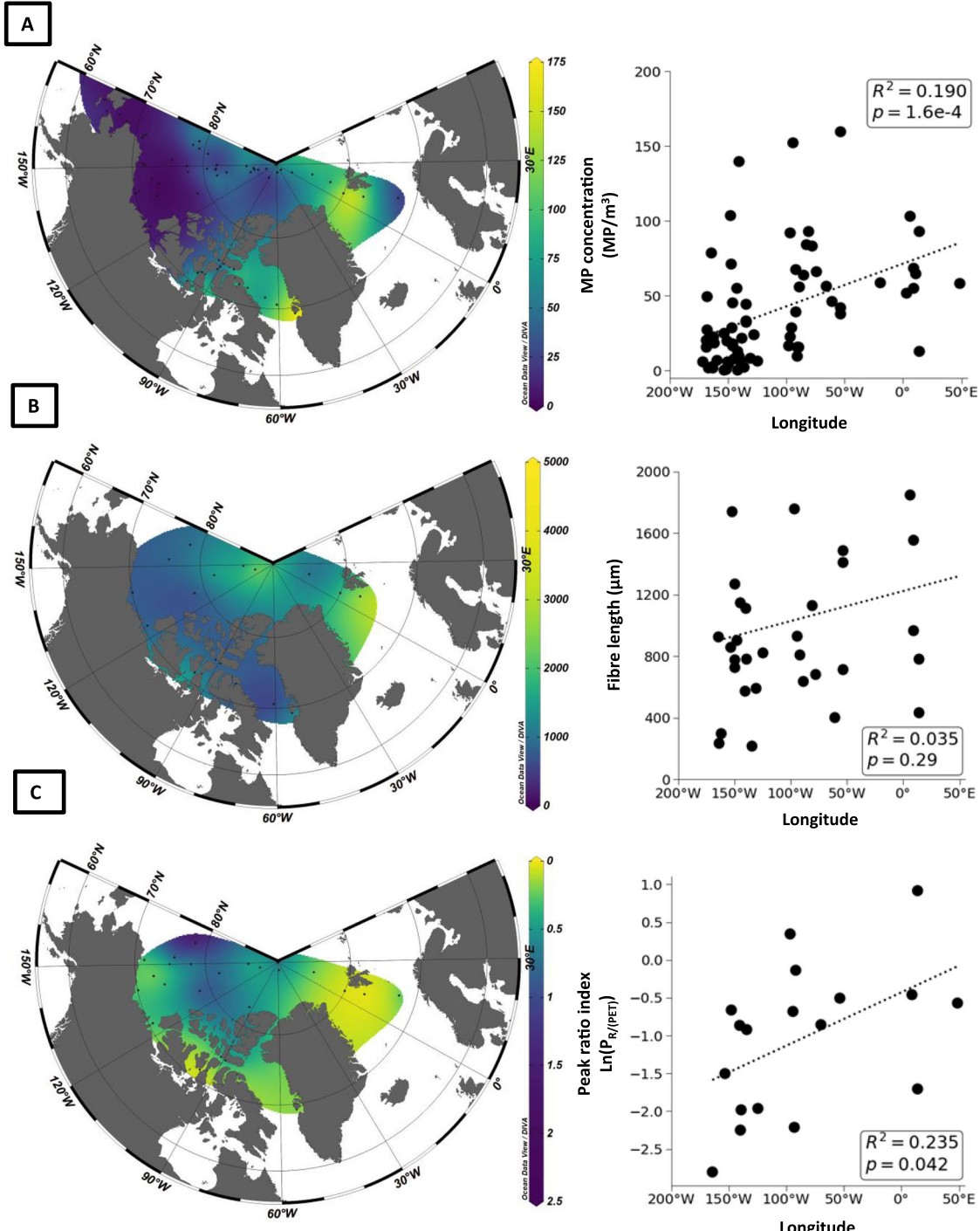

**Fig. 2 Microplastic concentration, fibre length, and polyester profiles.** Contour plots and longitude regression analyses for **A** FTIR-confirmed microplastic concentration (MP m$^{-3}$); **B** mean fibre length (μm); and **C** Infra-red peak ratio indices for polyester fibres (index of the ratio of heights of a de novo peak and a commercial reference peak observed in infra red spectra for polyester; see section "Methods") across the Arctic Ocean. Data based on 71 near-surface (3–8 m depth) samples collected via vessel seawater intakes from four cruises aboard three vessels in summer 2016. Plots on the right panels show MP concentration, mean fibre length, and infra-red Peak Ratio Index (Ln transformed) as a function of longitude across the Arctic (regression data exclude samples from the polar region > 85ºN). Centre point is the North Pole, North America lies to the SW and Europe to the SE of each map. Ocean Data View 5.2.0. The raw data underlying the figures can be found in the Supplementary Table 2.

add to the source identification of Arctic MPs, we suggest that the combined, historical release of wastewater from Europe, the Americas and Asia, warrants additional scientific scrutiny but provide for immediate best practices and management interventions.

The prevalence of MP fibres[2,6,7] and polyester[4,6] in our samples is consistent with recent studies of near-surface water in the central Arctic Basin[2] and Hudson Bay and the eastern Canadian Arctic[8], but contrasts observations of polymer composition in Arctic sea ice and ice floes where varnish (polymer used

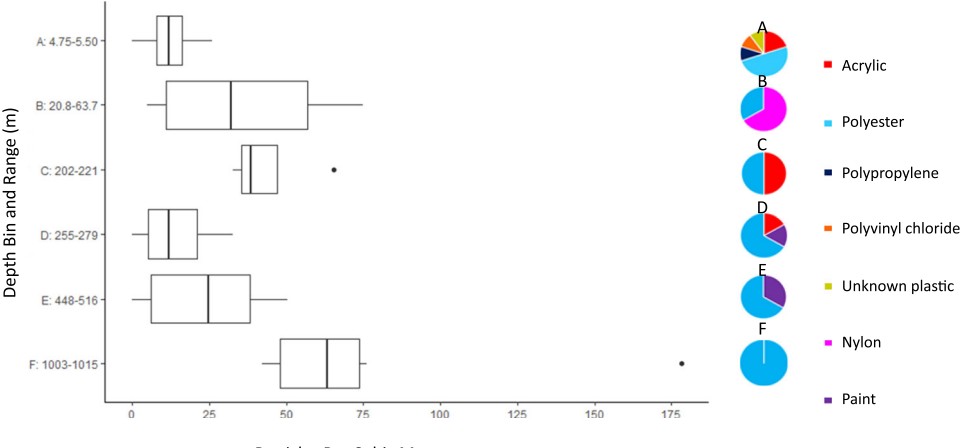

**Fig. 3 Microplastics were found throughout the water column in the Beaufort Sea.** Moderate microplastic (MP) counts were detected in the fresher surface waters that are a mix of Pacific-origin and Atlantic-origin water, river runoff and ice melt (**A**, salinity ~ 27 to 32.5 kg m$^{-3}$). Higher MP counts were detected in the core of the Pacific-origin water masses (**C**, salinity ~ 33 kg m$^{-3}$) and in the core Atlantic-origin water (**E**, salinity ~ 34.8 kg m$^{-3}$). Intermediate count water masses (**B**, **D**, **F**) are approximately summer-formed Pacific-origin water (**B**), the transition from Pacific-origin to Atlantic-origin water (**D**) and deeper Atlantic-origin water (**F**). Polyester dominated the profile of MPs at all (**A**, **C**, **D**, **E** and **F**) but one (**B**) depth sampled. Combined data of 26 samples collected at six Beaufort Sea stations. Each boxplot indicates the median (central mark) and 25th and 75th percentiles, with whiskers outside the box indicating 10th and 90th percentiles. Points above and below the whiskers indicate outliers.

in coatings)[5,21], and polyethylene and polypropylene fragments[5] were found to be abundant.

Our results show a clear correspondence between low MP counts and what appear to be more weathered fibres in the Pacific-influenced western Arctic (Bering Sea, Chukchi Sea, Beaufort Sea and Beaufort Gyre), and high counts of less-weathered fibres in the Atlantic-influenced eastern Arctic. The North Pole region is a transition between the east and west. Following this correspondence between MP fibres and Arctic water masses, we suggest that relatively large quantities of fibres are entering the Arctic from the North Atlantic, while fewer and older fibres in the western Arctic may result from smaller (and potentially older) inputs from the Pacific and Atlantic Oceans.

The large North American and Eurasian river inflows to the Arctic Ocean must remain in consideration as potential sources of MP fibres, as our data show only a broad variation between the Atlantic Arctic and Pacific Arctic. In addition, the potentially important role of atmospheric delivery of MPs into the Arctic remains a much-needed area of research. The dominance of polyester fibres in our study underscores the potentially important role that textiles, laundry and wastewater discharges may have in contaminating the world's oceans with MPs. In this regard, the inherent vulnerability of the remote Arctic is notable.

## Methods

**Sample collection.** Near-surface microplastic samples were collected via seawater intake loops at 71 stations aboard four scientific cruises in 2016: the Distributed Biological Observatory (DBO, IOS cruise number 2016-17) aboard the *CCGS Sir Wilfrid Laurier* (*SWL*) from 1 to 21 July (with samples from the North Pacific, Bering and Chukchi Seas); the United Nations Convention of the Law of the Sea (UNCLOS, IOS cruise number 2016-15), aboard the *CCGS Louis S. St Laurent* (*LSSL*) from 7 August to 18 September (with samples collected along a transect from Tromsø, Norway, passing over the North Pole and into the northern Canada Basin); the Joint Ocean Ice Study (JOIS, IOS cruise number 2016-16), also aboard the *CCGS LSSL* from 22 September to 18 October (with samples from the Canada Basin), and aboard the One Ocean Expeditions *RV Akademik Ioffe* (*OOE*) from 14 to 23 August (with samples from western Greenland to the Canadian Arctic Archipelago) (Fig. 1).

All near-surface seawater samples (3–8 m depth) were collected while the vessels were in transit, navigating between 5 and 18 knots, slowing down when transiting through ice. Stainless steel piping took the water from the seawater intake to the sampling laboratory. Seawater samples were sieved through a 63 μm brass sieve (WS Tyler USA Standard Test Sieve No. 230, 63 μm) as previously described[58], with volumes (recorded) averaging 68.6 L between programmes

(28.6–460.3 L). Particles retained on filters were rinsed with MilliQ water into 50 mL glass vials and stored at 4 °C until processing at the Ocean Wise Plastics Lab in Vancouver, BC.

In addition to the near-surface samples, 26 water column profile samples were collected at six locations using CTD/rosette equipped with 24 × 10 L Niskin bottles. In these cases, 29–67 L of seawater was sieved using the protocol above. Depth profile samples were collected through the JOIS programme by the *CCGS LSSL*. Data were normalized to cubic metre and reported as total suspect MP per station sampled or as average microplastic count ± standard error (SEM) in the Arctic water mass (Supplementary Table 2).

Each *CCGS* vessel was equipped with a Seabird SBE21 thermosalinograph (TSG) measuring seawater-loop temperature and salinity. The CTD/Rosette had a Seabird SBE911 + CTD system measuring the same parameters in addition to depth. Data were processed using standard routines in the Seabird data processing software (SBEDataProc). TSG data were processed into 30-s bins and CTD data into 1-dbar bins.

**External contamination controls.** MP analyses of environmental samples are sensitive to air-borne MPs, with microfibres being particularly abundant in indoor air[59]. Yet, there exist no universally accepted methods to account for field or lab-based contamination of samples with MPs, with approaches varying between studies. In our study, we applied stringent measures to reduce and account for microplastic contamination from field and laboratory procedures. In the field, air blanks were collected alongside sampling. Preparation of field samples for micro-plastic analysis was carried out at an isolated laboratory, which receives purified air via high-efficiency particulate air (HEPA) system. The laboratory enforces regular cleaning regime and contamination is regularly monitored for any systemic air-borne MPs using air blanks. At the time of study cotton lab coats were employed by all researchers operating in the lab. Any metal tools and glassware used for sample processing and analyses were rinsed three times with water filtered through a 1 μm borosilicate glass fibre filter. During processing all equipments, including funnels and containers for samples were covered with tinfoil and exposed to air as little as possible to prevent incidence of external contamination. Further, all handling of samples was carried out in a biological safety cabinet (BSC) or the HEPA laminar flow bench.

Microplastic external contamination was assessed by means of filtrate blanks (field), air blanks (field and laboratory) and procedural blanks (laboratory). The air blanks were used to assess any systemic contamination present in the environment during field sampling and sample laboratory processing. This involved exposing a 47 mm GF/F filter in the vicinity of sample collection and processing area. After sample processing, the filters were folded and stored in aluminium pockets. Procedural blanks were collected during sampling (1 in every 10 samples, Supplementary Table 3). The number of fibres in procedural field filtrate blanks ranged from 4 to 20, with an average of 9 ± 5.42, with the most common type being orange fibres, likely from cotton laboratory coats.

To account for any artifacts arising from microplastic contamination in our field samples, particles that were visually similar and present across multiple samples were removed from the final analysis. This conservative approach applied in previous research[4] resulted in a large number of particles excluded (a total 60%),

but is validated by the similarity of our final estimates to others made for the region[2]. There were no differences among the microplastic contamination values on different cruises and vessels (ANOVA, $p > 0.05$).

**Sample preparation for microplastic analysis**. MPs were extracted using a density-independent oil extraction methodology as described elsewhere[60]. An ethanol (95%) soak was applied to remove any oil coating the surface of MPs to avoid interference with the subsequent infra-red analysis. Extracted particles were vacuum-filtered through 10 μm-pore-sized polycarbonate filters inside the Laminar Flow Hood in preparation for microscopy visual assessment and FTIR spectro-metry identification of microplastic polymer types.

**Microplastic enumeration**. Quantification of microplastic loads was achieved with a two-step analysis: (1) visual microscopy identification of suspect microplastics (SMP) in each sample and (2) a single point μFTIR analysis to confirm that SMPs were indeed MPs, and for polymer identification in a subset of SMPs from every near-surface and depth-related sample (37.6%). The use of focal-plane array detectors in FTIR analysis (FPA-FTIR) offer the opportunity to perform polymer source imaging on filter areas or whole filters without the need of pre-sorting of MPs under the light microscope[4,61]. While FPA-FTIR offers considerable potential to enhance throughput in microplastic analysis, it remains a recent development[62]. Further, biofouling material or other organic matter coating MPs in samples, as we encountered in some cases in the present study, can affect the quality of infrared spectra of polymers, necessitating particle washing and re-analysis on FTIR to reliably confirm microplastic identity.

Filters were surveyed for suspect MPs under a stereo microscope (Olympus SZX16 microscope with Olympus DP22 camera, Tokyo, Japan) coupled to an image analysis software (DP2-SAL software, Olympus, Tokyo, Japan). Plastic selection criteria were adopted from published research[63,64]: (i) the particle contains no visible cellular structures; (ii) the fibre has a constant width and even coloration; (iii) the ends of the fibre are flat and not tapered to a point or frayed; and (iv) the fibre curls, crimps, or bends in three dimensions, and can stand partially upright on the filter or microscope slide. A separate set of visual criteria was adapted for non-fibrous particles[62,63] that included: (i) the particle has sharp, relatively straight edges and even colouration and (ii) the particle does not easily deform or break apart when poked with a fine needle.

Particles identified visually were categorized as (i) not MPs (N) when meeting none or only one of the criteria; (ii) uncertain (U) when meeting two of these criteria, or weakly fulfilling them; or (iii) suspected microplastic (Y) when meeting three or more criteria.

As a particle's fulfilment of the MP criteria was a subjective judgement[27], the success rate of plastic identification for an individual microscope operator was calculated using the results of Fourier-transform infrared (FTIR) spectroscopy, coupled with a power analysis to ensure sufficient sample size. The data to inform this procedure was obtained via FTIR analysis of all SMPs in one in every four of the near-surface water samples and all non-contaminant particles from depth samples, with additional particles chosen from some near-surface samples initially for method validation ($n = 20$ surface samples, total of 590 [37.6%]). Our method of selecting particles for FTIR analysis was designed to conserve instrument time, while providing a statistically powerful validation of the proportion of plastic particles counted with microscopy. A total of 590 of 1570 (37.6%) SMPs scanned with FTIR allows an estimation of the composition of the total population of 1570 particles with a confidence of 95% and a margin of error under 10%. This suggests that our error analysis can be confidently applied to enumerate MPs in samples where no FTIR was applied. The validation procedure included particles in air and procedural blanks to ensure suspect contaminants excluded from the analysis were of the same material, which, in combination with visual similarities was assumed to indicate field contamination. Blank data are presented in Supplementary Table 3.

In our validation dataset, which includes all particles later flagged as contaminants, 10.7% of N particles, 18.2% of U particles, and 39.8% of Y particles were plastic, underscoring the importance of post-hoc, non-visual means to identify plastic polymers and correct concentration calculations. This provided a basis for the weighted average calculation for samples where particles were not assessed under FTIR to correct for any uncertainty associated with visual microscopy defined here. For depth samples, a full FTIR assessment was carried out.

Weighted estimate of the total number of MPs in a sample that accounts for the rate of particles false-positively identified as plastics and uncertain particles, with the data normalized to a cubic metre of seawater:

$$\text{MP m}^{-3} = \frac{\left(k_{\text{confirmed}} + k_N \text{rate}_N + k_U \text{rate}_U + k_Y \text{rate}_Y - l_{\text{nonplastic}}\right)}{\text{volume}} \times 1000,$$

where in an individual sample, $k_{\text{category}}$ is the number of MPs in each category, $\text{rate}_{\text{category}}$ is the percentage of each category that were confirmed plastic, and $l_{\text{nonplastic}}$ is the number of scanned particles that were not plastic (including natural particles struck from the count), and volume is in liters.

**Fourier transform infrared spectrometry**. Suspected MPs were analysed using a Cary 670 FTIR spectrometer equipped with a Cary 620 microscope (Agilent

Technologies, Mulgrave, AUS). Particles, including individual microfibres, were analysed using the micro-ATR accessory equipped with a Germanium crystal. For this, individual particles were affixed to a glass microscope slide covered with a thin layer of 2% dextrose (Sigma-Aldrich, St. Louis, USA) using microtweezers. For each sample, 128 co-added scans at a resolution of 8 $\text{cm}^{-1}$ in the range of 3800–900 $\text{cm}^{-1}$ were collected. The spectra were matched against commercial library of FTIR spectra (KnowItAll, Bio-Rad). Sample spectra were identified successfully if they met the following criteria: (i) all major peaks were present in both reference and sample spectra and (ii) the total overlap of the reference and sample spectra was >80%.

**Infra-red signatures of MPs across the Arctic**. Infra-red spectra of MPs pro-vided a basis for polymer identification, but the imperfect matches between spectral properties of environmental MPs with candidates in the reference libraries are a challenge. While some of these mismatches may be explained by incomplete nature of commercially available reference libraries, weathering of particles in the envir-onment also alters infrared spectra of plastic particles[25,28]. To characterize spectral alterations in response to weathering, we conducted a controlled pilot study of new materials of known composition that were weathered over the course of a year. Based on the resulting changes in infra red signatures, we calculated a Peak Ratio Index for two major peaks (one that varied and one that was stable) to support the characterization of spectral alterations of MPs across the Arctic.

Polyester fibres dominated our Arctic samples and therefore afforded us a large dataset to evaluate their spectral properties in the environment. The most prominent variation (i.e. increase in absorbance) for spectra of polyester materials in our controlled weathering study was in the region 950–1100 $\text{cm}^{-1}$, with maximum absorbance near 970 $\text{cm}^{-1}$ (assigned to C–H bond in vinyl group)[60]. We then derived a peak ratio index for polyethylene terephthalate (PET) from the ratio of heights of two peaks, at 970 $\text{cm}^{-1}$ as numerator and 1241 $\text{cm}^{-1}$ as denominator (Supplementary Fig. 1). The peak at 1241 $\text{cm}^{-1}$ was selected as a reference peak due to consistency of its height in our pilot spectral study of new and weathered PET. We do note that one of the other study qualitatively noted that solid polyethylene bottles exhibited less stability for the 1241 $\text{cm}^{-1}$ peak[34], underscoring the distinct need for a large database and additional studies on weathering of plastics and microplastic fibres in the environment.

To assess spatial trends in peak IR ratio indices across the Arctic in near-surface seawater samples (3–8 m depth), indices of all polyester fibres from individual stations were averaged and mapped.

## Data availability

All data generated by this study are included in the figures, tables and supplementary materials, or are available from the corresponding author.

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

## Acknowledgements

The authors are grateful to the crews of the *CCGS Wilfrid Laurier, CCGS Louis S. St Laurent,* and the One Ocean Expedition *RV Akademik Ioffe.* The authors gratefully acknowledge the financial and logistical support of the Department of Fisheries & Oceans, and its National Contaminants Advisory Group. The authors are grateful to

Catherine Lawton, Aaron Lawton and the One Ocean Expeditions onboard team for expert logistic support. The authors are indebted to John Nightingale for his Arctic support and leadership. The authors thank Amber Dearden and Mathew Watkins for assistance with manuscript preparation.

## Author contributions

P.S.R. conceived of the study and led the overall interpretation of results and drafting of the manuscript. S.C. performed microscopy and data analysis. E.V. oversaw the FTIR data interpretation and controlled weathering evaluations. A.E. operated the FTIR. S.P. analysed FTIR data. S.-A.Q. collected samples aboard research vessels. S.Z. collected samples aboard research vessels. E.S. collected samples aboard research vessels. J.E. collected samples aboard research vessels. A.M.P. contributed to data interpretation, Quality Assurance and manuscript preparation. B.W. oversaw CCGS expedition planning and oceanographic interpretations.

## Competing interests

The authors declare no competing interests.
