## [Peer Review File · Nature Communications]

Reviewers' comments:

Reviewer #1 (Remarks to the Author):

Review of "Arctic microplastics: polyester fibre profiles point to Atlantic origins" by Ross et al

In this manuscript, the authors present data on 71 stations in the (mostly Canadian) Arctic where they sampled microplastic fibres from ship water intake to analyse the distribution and possible sources of these fibres that they found. These are very valuable samples from an inaccessible place, and it is especially good to see that the different expeditions were all performed in the same year, creating a synoptic view of microfibrils across a large region. I do have some major comments:

1) I am not an expert in field and lab sampling of microplastics, but I know that for a long time colleagues have shied away from analysing fibres because the risk of contamination was so large. The focus on these fibres in this manuscript therefore really make me wonder how clean the sampling and analysis of the data here was; and I think the authors should put this much more upfront. While they do have a paragraph in the methods section about accounting for external contamination, they do not give any quantitative data on the number of fibres on their blanks. How much contamination was there?

2) In general, I find it hard to exactly understand their method. It is only at the end of the Methods section that the authors admit that they didn't use FTIR on all their samples, but only on 27% of them. Also nowhere does it state how many particles were found on each station. I suggest that the authors add these raw numbers (both visual number of SPM and number analysed in FTIR) in their Table 1, for completeness.

3) The authors mention that their microplastic concentrations are likely much higher than previously reported because they filter to much smaller sizes. But they could check this statement by redoing the analysis for only the larger particles. Do they then get to values closer to that in previous studies?

4) Sampling from the ship's water intake is not very commonplace in the community; most sampling is done by neuston net trawling so that the water surface is skimmed. The authors should relate their findings to the more traditional sampling methods to facilitate comparison, or at least more clearly point out the differences?

5) The ODV maps of Figure 2 rely far too much on interpolation. With only so few data points, and well-documented fine-scale variability in plastic abundance, simple interpolation across the entire Canadian Arctic is uncalled for. I suggest the authors change the maps in this Figure to highlight the values at the sampling points (using e.g. colour-coded circles)

6) The discussion of depth profiles assumes that the fibres are passively transported in 3D by the ocean currents. However, this completely ignores the possibility that some of the fibres/particles have or are sinking locally. In combination with the local stratification, this can also give rise to variations in concentration with depth. The authors should thus be much more careful with their interpretation, and for example use numerical simulations to back up their hypotheses on the 3D flow

Minor comments:

- Figure 1: The straight arrows for Atlantic and Pacific inflow are far too simplistic.
- Figure 1: Also show the locations of the deep sampling in the Beaufort gyre on this map?
- Line 86: Here, it would be good to mention that only 27% of samples are FTIR analysed
- line 96: why does this underscore the need? Is there any known impact these fibres?
- line 124: Since sea water salinity co-varies strongly with longitude, I am doubtful that this adds any more information than the correlation with longitude itself already does. I suggest the authors remove this analysis

- line 214: 3.5e15 microfibrils sounds like a large number, but since there is 1.3e18 m³ of water in the ocean, this is only one fibre every 400 m³ of water. Even if all these fibres would end up in the Arctic (which is 1% of the ocean value), that is less than 1 fibre per m³. So this back-of-the-envelope calculation cannot at all explain the findings here, and thus I suggest the authors remove this suggestive statement.

- Line 243: What is the potential role of fisheries in generating (polyester) plastic fibres/particles? It is surprising that the authors mention nothing about that. Do they have a conflict of interest?

- Line 436: I do not understand how the power analysis was conducted. A bit more detail here would be useful. For examples, are 'Y' particles not counted twice (once in k_confirmed and then again in k_Y*rate_Y)?

Type-os and grammar:

- Line 16: word 'functions' is redundant here

- Line 16: 'The Arctic is not immune' is an anthropomorphism

- Figure 1A, right panel: minus sign missing in $p=4.89e06$?

Reviewer #2 (Remarks to the Author):

The manuscript "Arctic microplastics: polyester fibre profiles point to Atlantic origins", submitted to Nature Communications, is addressing the issue of a remote and fragile ecosystem, the Arctic Ocean, being influenced by microplastics pollution. The study covers the spatial distribution of microplastics in the Arctic horizontally and vertically.

One important feature of this study is the large dataset analysed. Another special feature is the focus on the physical properties of the microplastics identified, like the polymer type, shape and weathering profile. Next to a large dataset on surface water samples (71 samples) the study also covers a depth profile of the water column (26 samples), which adds some important information to a so far not very well investigated compartment.

The authors considered many recent publications for this manuscript and connect their results well to the recent state-of-the-art in the field of microplastics research in the Arctic.

Overall, the manuscript is well written and enjoyable to read and the figures included in the manuscript nicely illustrate the main findings of this study.

My biggest point for criticism is the analytical approach. The study relies on visual inspection and identification of microplastics. However, the authors performed a validation of a subset (26.9%) of the putative microplastics by using FTIR spectroscopy, a widely used and acknowledged method for microplastic identification, and stressed the need to perform this kind of quality assurance. It becomes not quite clear how this subset was decided for (e.g. randomly, based on certain returning criteria etc.)? I am also missing a comment on the use of different techniques less prone to human-bias, like μ FTIR-imaging, used in the cited publication by Bergmann et al. 2019 (doi: 10.1126/sciadv.aax1157). Furthermore, I am missing a note on other QA/QC procedures like laboratory blanks, since contamination, especially with fibres, is considered an issue in microplastics research (e.g. Wesch et al 2017 (doi: 10.1038/s41598-017-05838-4)). I am aware that this has been addressed to some extent in the method section, but a short discussion would be preferable to have also in the main text since it is a crucial point of this study. Furthermore, the results for the analysed blanks could be provided in the extended data section.

The aspect focussing on the weathering profile of PET fibres is very interesting and of importance, since data on environmentally weathered microplastics is rather scarce. However, there has been a study published by Ioakeimidis et al. 2016 (doi: 10.1038/srep23501) which identified different bands signalling for degradation. Noteworthy, one of the bands they found decreasing with increasing weathering was at 1245 cm⁻¹, which was considered as steady reference band in this study. A short statement/discussion in the context of the study by Ioakeimidis et al. 2016 would be highly appreciated. It is also not apparent how it was ensured that the changes in FTIR spectra, which serve the determination of the weathering index, are not a consequence of the sample treatment (e.g. have particles been sufficiently dried after treatment before measurement since FTIR spectroscopy is very

sensitive to remaining water) or result of baseline effects e.g. has a normalization or baseline correction been applied to the spectra)?

In the abstract, the importance of the Arctic as ecosystem also in the context with food safety and the Indigenous people is discussed. However, this is not discussed further in the manuscript but should be considered.

Specific comments by line number:

L30: The authors define "microplastics" as "MP" but continue to use the abbreviation "MPs" in the following text. Thus, I would suggest to change the first abbreviation to "MPs".

L32: A very recent study focussed on microplastics in Arctic surface water, sediment and the water column, which might be worthwhile to be included here: Tekman, M.B., Wekerle, C., Lorenz, C., Primpke, S., Hasemann, C., Gerdt, G., Bergmann, M., 2020. Tying up loose ends of microplastic pollution in the Arctic: Distribution from the sea surface, through the water column to deep-sea sediments at the HAUSGARTEN observatory. *Environ. Sci. Technol.* doi: 10.1021/acs.est.9b06981

L60-64: This paragraph, highlighting the aim of this study, is not very well integrated into the preceding part of the introduction and should be connected better to the before mentioned gaps of knowledge (e.g. that not much data exists on the water column especially in the Arctic Ocean). Furthermore, the weathering aspect mentioned before could be addressed here as well.

L89-91: The results of this study fall however right in the range of the recently published study by Tekman et al. 2020 (doi: 10.1021/acs.est.9b06981), which should be mentioned in this context. Tekman et al. 2020 used 32 µm filters for sampling being therefore comparable to the 63 µm mesh used in this study. While Tekman et al. 2020 used the superior analysis technique of µFTIR imaging, they were unable to clearly differentiate between fibres and particles which is an advantage of this study.

L110-118: Information is missing on how many datapoints (n), being confirmed microplastics, fibres, PET fibres, were considered. The authors only state the number of samples considered but do not state here whether data were taken from individual particles/fibres and if so how many or if averaged etc. per sample.

Furthermore, I short remark on the scale of the weathering index (e.g. the higher the value the more weathered the PET fibre?) would be preferable. It is also a bit irritating why this scale is opposed to the other two, namely having the high values in the violet colour section.

L126: "of" missing: "This could reflect lesser inputs of MPs..."

L138: The following reference should be included: Ioakeimidis et al. 2016 (doi: 10.1038/srep23501)

L147: Just one "l" for "Kruskal-Wallis"

L156: Maybe the relative size of the analysed subset (26.9%) could be stated here as well

L168: It is not quite clear if the term "water-mass related" here is referring to a certain water body or the actual mass of particles/sample etc. I guess it refers to "water mass" and the hyphen should be erased or the term water body used.

L168-177: Apart from the higher microplastic concentration in the surface water it is also interesting to point out that the polymer composition shows a larger variety compared to the deeper water masses. One more remark with regard to Figure 3 would be the composition of "paint". Is it mainly polyurethane-based or acrylic-based paints?

L219-220: In the study by Tekman et al. 2020 polyester was among the six most dominant polymer types.

L219-223: The here mentioned publications by Bergman et al. 2019 and Peeken et al. 2018 also focus on small sized microplastics (down to 11 µm). Thus, the differences in the reported polymer types are more likely to be explained by the different analysis techniques (visual inspection + FTIR vs. µFTIR imaging) and the different reference databases used.

L248-254: The first three references have a different reference style than the others. This should be coherent.

L292-294: same comment on incoherent reference style

L380: It is a bit unclear whether the total number of samples analysed was 71 or whether this is the number of analysed surface water samples and the 26 water column samples were additional samples.

L391-401: In the light of the relatively low microplastic concentrations (102-103 particles m⁻³) it

should be discussed how representative sample volumes of on average 0.07 m³ are.

L437-438: Does this mean that for all the particles sorted as "not microplastics" in this study, which were further analysed using FTIR spectroscopy, 10.7% were actually plastics and so on?

L440-445: Unfortunately, I cannot really follow the calculation of the microplastic concentrations since the variables used are not all defined clearly. Maybe it would be a bit more transparent if the data were provided as a table etc. It is also a bit misleading that concentrations labelled "FTIR-confirmed" seem to be estimations based on the actually confirmed microplastics + the number of suspected and not suspected micropastics that "should be" also microplastics based on identification rate? If this is the case, this needs to be made clear in the results/discussion!

L488: What does "ample dataset" mean? What is the number of fibres/spectra used for this approach?

L491: An additional and fitting reference could be Tidjani 2000 ([doi.org/10.1016/S0141-3910\(00\)00039-2](https://doi.org/10.1016/S0141-3910(00)00039-2)).

L500: remove one "seawater"

L516-518: different reference style

L562: "microplastics" should be abbreviated as "MPs" for coherence. Does the term "sample" refer to the actual number of samples or particle/fibre?

Reviewer #1 (Remarks to the Author):

Review of “Arctic microplastics: polyester fibre profiles point to Atlantic origins” by Ross et al

In this manuscript, the authors present data on 71 stations in the (mostly Canadian) Arctic where they sampled microplastic fibres (*we did not just sample microfibrils - we captured all microplastics, finding that most were fibres*) from ship water intake to analyse the distribution and possible sources of these fibres that they found. These are very valuable samples from an inaccessible place, and it is especially good to see that the different expeditions were all performed in the same year, creating a synoptic view of microfibrils across a large region. *We thank the reviewer for appreciation just how much work went into the sampling (four icebreakers in one field season as this laid the foundation for a strong interpretative analysis).*

I do have some major comments:

1) I am not an expert in field and lab sampling of microplastics, but I know that for a long time colleagues have shied away from analysing fibres because the risk of contamination was so large. The focus on these fibres in this manuscript therefore really make me wonder how clean the sampling and analysis of the data here was; and I think the authors should put this much more upfront. While they do have a paragraph in the methods section about accounting for external contamination, they do not give any quantitative data on the number of fibres on their blanks. How much contamination was there?

We agree that fibres are challenging as the potential for contamination in field and lab efforts are great. However, far from ‘shying away from’ analysing fibers, the current international microplastics science community is delve aggressively into fibers, based on the increasing realisation that fibres appear to be dominating the spectrum of microplastic types in water. In addition, we remain highly confident that our practices and procedures have reduced to zero or near zero the risk of contamination of samples by fibres in the field and in the lab. Over the 8 years that our specialised microplastics laboratory has been operating, we have developed and validated strong Standard Operating Protocols (SOPs) to address the potential for contamination during field sampling, laboratory sample clean up, and analysis, and to pinpoint artefacts of contamination.

Line 470-485 details our blank results. We have also added a new Table with raw data for blanks in the supplementary material (Extended Data Table 2)

Protocols that we have developed and applied to minimise contamination at every step include:

- *rules on clothing allowed while working;*
- *cleaning of all sieves, sample containers and other supplies with filtered water;*
- *clean air environments in the lab;*
- *regular surface cleaning (counter/floors/instrument surfaces);*

- *Standard Operating Protocols (SOPs) that cover sampling, extraction, analysis, measurements and researcher conduct (clothing and cleaning).*
- *In addition, the following steps inform our subsequent data analysis as to the potential for contamination:*
- *collection of samples of fibres from staff clothing when conducting field sampling;*
- *field and lab air and procedural blanks;*
- *analysis of the full dataset for anomalies that could be explained by cruise or individual-based contamination.*

2) In general, I find it hard to exactly understand their method. It is only at the end of the Methods section that the authors admit that they didn't use FTIR on all their samples, but only on 27% of them. Also nowhere does it state how many particles were found on each station. I suggest that the authors add these raw numbers (both visual number of SPM and number analysed in FTIR) in their Table 1, for completeness.

Both reviewers comment on aspects relating to Methods employed – which is encouraging, since comparable (or standardized) methods across different studies have become an important element of scientific discourse and inter-laboratory evaluation in the area of microplastic pollution research. We have done three things to improve the clarity and the detail to our Methods.

First, we have added a paragraph in the main body of the text to provide a high-level snapshot of the methods we used. We agree that without this, the average reader would miss several key elements of the methods we designed, adapted or employed in our study. This is a compromise. We acknowledge that the format of the journal is to place Methods into Supplemental materials. We hope that by having one short summary paragraph, readers will benefit from a basic understanding of what we did in the main body of the text, and subsequently can seek detailed information in the Methods.

Second, we thoroughly revised the Methods section to elaborate on the steps taken and respond directly to the specific questions posed by the reviewers (Line 496-551). We provide an Extended Data Table 2 which includes data on blanks.

Finally, we have added total Suspected microplastic counts and total number scanned to inform the extent to which individual particles were scanned using FTIR (two new columns in Table 1).

3) The authors mention that their microplastic concentrations are likely much higher than previously reported because they filter to much smaller sizes. But they could check this statement by redoing the analysis for only the larger particles. Do they then get to values closer to that in previous studies?

Our results show a preponderance of fibres that can easily pass through different sieve sizes in the field – their diameter typically ranges from 7 to 14 microns. A recent study in Hudson Bay and related Arctic waters found values very close to ours (added to our paper; Huntington, A. et al. A first assessment of microplastics and other anthropogenic particles in Hudson Bay and the surrounding eastern Canadian Arctic waters of Nunavut. FACETS 5, 432–454 (2020), which provides independent support for our observations.

4) Sampling from the ship's water intake is not very commonplace in the community; most sampling is done by neuston net trawling so that the water surface is skimmed. The authors should relate their findings to the more traditional sampling methods to facilitate comparison, or at least more clearly point out the differences?

We deliberately designed a study which would avoid the bias towards lower density particles (studies using Manta surface sampling find large amount of expanded polystyrene), and we were interested in the biota-inhabiting (pelagic) water column. Further, sampling with manta trawls excludes smaller sized microplastics as the porosity of their meshes is typically in the range of 200 - 333 μm . Sampling from the intake represents one of the several approaches in this evolving field (e.g. Enders et al. 2015, Desforges et al. 2015, Cincinelli et al. 2017; Arctic studies: Kanhai et al. 2018, Lusher et al. 2015). There is also some evidence that ocean surface measurements underestimate microplastic abundance (Kukulka et al. 2012).

We have referred to those we consider as most relevant you our study:

- *Kanhai, L. D. K. et al. Microplastics in sub-surface waters of the Arctic Central Basin. Mar. Pollut. Bull. 130, 8–18 (2018).*
- *Lusher, A. L., Tirelli, V., O'Connor, I. & Officer, R. Microplastics in Arctic polar waters: The first reported values of particles in surface and sub-surface samples. Sci. Rep. 5, 14947 (2015).*
- *Desforges J-PW, Galbraith M, Dangerfield N, Ross PS. Widespread distribution of microplastics in subsurface seawater in the NE Pacific Ocean. Mar Pollut Bull. 2014;79: 94–99. pmid:24398418*

5) The ODV maps of Figure 2 rely far too much on interpolation. With only so few data points, and well-documented fine-scale variability in plastic abundance, simple interpolation across the entire Canadian Arctic is uncalled for. I suggest the authors change the maps in this Figure to highlight the values at the sampling points (using e.g. colour-coded circles)

We designed this comprehensive figure to do two things: i) provide an easy to understand high level overview of the spatial distribution of microplastics across the Arctic (the colour ODV maps) and ii) to match these alongside statistical /data driven plots that provide insight into the underlying processes. We acknowledge the reviewer's point here, but would very much prefer to keep as is. While a larger sample size would be most helpful, it is not realistic

– collecting samples aboard four vessels from 71 stations is a massive undertaking. We feel that this duality of figures – one more descriptive plus the other statistical provides the reader with a balanced and defensible view of our findings.

6) The discussion of depth profiles assumes that the fibres are passively transported in 3D by the ocean currents. However, this completely ignores the possibility that some of the fibres/particles have or are sinking locally. In combination with the local stratification, this can also give rise to variations in concentration with depth. The authors should thus be much more careful with their interpretation, and for example use numerical simulations to back up their hypotheses on the 3D flow

We agree with the reviewer and acknowledged the uncertain role of microplastic sinking in the initial submission. Much of the existing literature on microplastics focuses on the upper ocean, with less data available on the fate of particles at depth. However, emerging studies suggest that surface microplastics may be subject to downward transport through physical and biological processes (Chubarenko et al. 2017). Our intention was to explore the potential role of water masses in influencing the microplastic fate in the Arctic, rather than conclusively attribute their distribution to water mass transport.

Chubarenko I. Microplastics Migrations in Sea Coastal Zone: Baltic Amber as an Example. Fate and Impact of Microplastics in Marine Ecosystems. Elsevier; 2017. pp. 15–16

Minor comments:

- Figure 1: The straight arrows for Atlantic and Pacific inflow are far too simplistic.

The goal of this graphic is not to depict surface water circulation patterns in the Arctic (this is not a physical oceanography study and no such data were generated) but these arrows were to illustrate to the reader as to the possible underlying mechanisms explaining the distribution of microplastics observed in our study. Higher abundance and “fresher” microplastics were measured in the eastern Arctic that also experiences higher inflow of water. This contrasted the eastern region where the reduced inflows and connection to the older waters of the Pacific may help explain the lower abundances and more aged microplastics. We respectfully suggest keeping to this original depiction of broad circulation patterns in the Arctic, with supporting text in the Figure legend.

- Figure 1: Also show the locations of the deep sampling in the Beaufort gyre on this map?

Thank you for this suggestion. We updated Figure 2 with stations where samples were collected at depth (JOIS Depth Stations).

- Line 86: Here, it would be good to mention that only 27% of samples are FTIR analysed. *Done and corrected.*

- line 96: why does this underscore the need? Is there any known impact these fibres? *We feel that we have speculated appropriately throughout the manuscript and are simply*

stating that 'more research is needed' – perhaps one of the most common means to express a cautionary assessment in the scientific community □

- line 124: Since sea water salinity co-varies strongly with longitude, I am doubtful that this adds any more information than the correlation with longitude itself already does. I suggest the authors remove this analysis.

We agree with the reviewer and we removed this figure and corresponding interpretations from our manuscript.

- line 214: 3.5×10^{15} microfibrils sounds like a large number, but since there is 1.3×10^{18} m³ of water in the ocean, this is only one fibre every 400 m³ of water. Even if all these fibres would end up in the Arctic (which is 1% of the ocean value), that is less than 1 fibre per m³. So this back-of-the-envelope calculation cannot at all explain the findings here, and thus I suggest the authors remove this suggestive statement.

We feel that the evidence from our team and from other citable works provide compelling evidence of a growing textile-based source for microfibrils in the oceans. Given our identification of synthetic fibers (primarily polyester) that are structurally similar to textile fibres, the growing evidence of significant wastewater releases of microfibrils, we feel that our interpretation is a logical one. Our cited work is far from a back of the envelope calculation – it reflects our own published research into microplastics released from the largest wastewater treatment plant in British Columbia and an estimate for cumulative releases by all facilities in Canada and the USA. The reviewer acknowledges that the Arctic Ocean is vast – but fails to note that our estimate (cited) is for just one year, and captures only Canada and the USA – no European or Asian calculations. This means that the potential for wastewater sources to be much larger than the reviewer expects. Nonetheless, the reviewer comment points out an opportunity for us to clarify this point. We added: "While further inventories will no doubt add to the source identification of Arctic microplastics, we suggest that the combined, historical release of wastewater from Europe, the Americas and Asia, warrants additional scientific scrutiny and offers mitigation opportunities in the form of best practices and management interventions."

- Line 243: What is the potential role of fisheries in generating (polyester) plastic fibres/particles? It is surprising that the authors mention nothing about that. Do they have a conflict of interest?

Polyester fibers have their main application in textile making, whereas fishing nets and lines are made primarily with nylon (Li et al. 2016); in addition fishing industry can be a source of polypropylene and polyethylene fibers that are used in ropes as reported in Xue et al. 2020. We have modified to respond to this important reviewer comment.

Xue, B., Zhang, L., Li, R., Wang, Y., Guo, J., Yu, K., and Wang, S., 2020. Underestimated Microplastic Pollution Derived from Fishery Activities and "Hidden" in Deep Sediment.

- Line 436: I do not understand how the power analysis was conducted. A bit more detail here would be useful. For examples, are 'Y' particles not counted twice (once in k confirmed and then again in $k_Y \cdot \text{rate}_Y$)?

We hope that our modified methods section clarifies reviewer's comment (Line 496-551). The power analysis was performed on a subset of particles drawn from the entire dataset (all samples across the Arctic and at depth) and validates the broad trends in microplastic composition across our region (Extended Data Table 1). It was not applied to the equation, which represents a method to arrive at microplastic counts per sample (MP m⁻³) corrected for false-positive and negative errors, and the rate of counting uncertain particles (those that cannot be identified as either plastic or natural).

For this calculation, we performed a complete FTIR analysis of suspect microplastics in 1 in every 4 samples, and additional randomly chosen particles from the remaining samples. This amounted to 590 particles out of the 1570 that were used as a basis of identifying the error rates.

Type-os and grammar:

- Line 16: word 'functions' is redundant here. *Changed.*
- Line 16: 'The Arctic is not immune' is an anthropomorphism. *Changed.*
- Figure 1A, right panel: minus sign missing in $p=4.89e06$? *Good catch! Changed.*

Reviewer #2 (Remarks to the Author):

The manuscript "Arctic microplastics: polyester fibre profiles point to Atlantic origins", submitted to Nature Communications, is addressing the issue of a remote and fragile ecosystem, the Arctic Ocean, being influenced by microplastics pollution. The study covers the spatial distribution of microplastics in the Arctic horizontally and vertically. One important feature of this study is the large dataset analysed. Another special feature is the focus on the physical properties of the microplastics identified, like the polymer type, shape and weathering profile. Next to a large dataset on surface water samples (71 samples) the study also covers a depth profile of the water column (26 samples), which adds some important information to a so far not very well investigated compartment.

The authors considered many recent publications for this manuscript and connect their results well to the recent state-of-the-art in the field of microplastics research in the Arctic. Overall, the manuscript is well written and enjoyable to read and the figures included in the manuscript nicely illustrate the main findings of this study.

My biggest point for criticism is the analytical approach. The study relies on visual inspection and identification of microplastics. However, the authors performed a validation of a subset (26.9%) of the putative microplastics by using FTIR spectroscopy, a widely used and acknowledged method for microplastic identification, and stressed the need to perform this kind of quality assurance. It becomes not quite clear how this subset was decided for (e.g. randomly, based on certain returning criteria etc.)?

We performed a complete FTIR analysis of suspect microplastics in 1 in every 4 samples, and additional randomly chosen particles from the remaining samples (Line 523-526). This amounted to 590 particles out of the 1570 that were used as a basis of identifying the error rates.

I am also missing a comment on the use of different techniques less prone to human-bias, like μ FTIR-imaging, used in the cited publication by Bergmann et al. 2019 (doi: 10.1126/sciadv.aax1157).

We have revised the Methods text to respond to this observation. MicroATR imaging has some promise, it also has some drawbacks – for example, small areas of a subsample is typically analysed as it is very time intensive. Our single point ATR offers much more confidence on the identify of SMPs and MPs.

Furthermore, I am missing a note on other QA/QC procedures like laboratory blanks, since contamination, especially with fibres, is considered an issue in microplastics research (e.g. Wesch et al 2017 (doi: 10.1038/s41598-017-05838-4)). I am aware that this has been addressed to some extent in the method section, but a short discussion would be preferable to have also in the main text since it is a crucial point of this study. Furthermore, the results for the analysed blanks could be provided in the extended data section.

Our modified methods section expands on the QA/QC procedures used in the study (Line 454-485) and we also attach blank data in the Supplementary Spreadsheet S1.

The aspect focussing on the weathering profile of PET fibres is very interesting and of importance, since data on environmentally weathered microplastics is rather scarce. However, there has been a study published by Ioakeimidis et al. 2016 (doi: 10.1038/srep23501) which identified different bands signalling for degradation. Noteworthy, one of the bands they found decreasing with increasing weathering was at 1245 cm^{-1} , which was considered as steady reference band in this study. A short statement/discussion in the context of the study by Ioakeimidis et al. 2016 would be highly appreciated. It is also not apparent how it was ensured that the changes in FTIR spectra, which serve the determination of the weathering index, are not a consequence of the sample treatment (e.g. have particles been sufficiently dried after treatment before measurement since FTIR spectroscopy is very sensitive to remaining water) or

result of baseline effects e.g. has a normalization or baseline correction been applied to the spectra)?

We acknowledge the good study by Ioakeimidis 2016, but do note that it refers to a study of polyethylene bottles, and not fibres. It also refers to a qualitative interpretation of IR signatures but does not advance a quantitative metric. We have built this reference into our text.

We developed a 'peak ratio index' after finding that our IR peak at 1245 cm⁻¹ varied little over time in a 365 day weathering study of commercial polyester and differed little from IR signatures found in our study; in contrast, the peak at 970 cm⁻¹ did change and it was on the basis of the data for these two peaks that we developed and applied the 'Peak ratio Index' plotted in Fig 2 c. We acknowledge the divergent experience in Ioakeimidis et al 2016 and suggest that further weathering studies remain a priority to clarify this point. In order to provide more transparent interpretation on this matter, we are clear that our weathering study remain at this stage a 'pilot' study, and we re-defined our index as 'Peak ratio index' rather than a 'weathering index'. This more cautious approach should allow better critical interpretation of our findings.

We are, in fact, in the midst of a study to address this question with 110 textile samples being exposed to the elements (air and seawater) and will be able to more substantively address the value of both approaches.

In the abstract, the importance of the Arctic as ecosystem also in the context with food safety and the Indigenous people is discussed. However, this is not discussed further in the manuscript but should be considered.

There have been no studies of microplastics in country foods of Inuit in the circumpolar region, but 'pollutants' have long been a priority concern in the North. We added to the Introduction a statement that does not speculate: "While data are scant, reports of microplastic in seafood have raised concerns about the potential for human ingestion and possible health effects"²³. For the indigenous peoples who rely heavily on foods from the ocean, including the Inuit inhabiting the circumpolar region, such concerns underscore the need for a more cohesive understanding of microplastic distribution and fate."

Specific comments by line number:

L30: The authors define "microplastics" as "MP" but continue to use the abbreviation "MPs" in the following text. Thus, I would suggest to change the first abbreviation to "MPs". *Done.*

L32: A very recent study focussed on microplastics in Arctic surface water, sediment and the water column, which might be worthwhile to be included here: Tekman, M.B., Wekerle, C., Lorenz, C., Primpke, S., Hasemann, C., Gerdt, G., Bergmann, M., 2020. Tying up loose ends of microplastic pollution in the Arctic: Distribution from the sea surface, through the water column to deep-sea sediments at the HAUSGARTEN observatory. *Environ. Sci. Technol.* doi: 10.1021/acs.est.9b06981.

Added, thank you.

L60-64: This paragraph, highlighting the aim of this study, is not very well integrated into the preceding part of the introduction and should be connected better to the before mentioned gaps of knowledge (e.g. that not much data exists on the water column especially in the Arctic Ocean). Furthermore, the weathering aspect mentioned before could be addressed here as well.

We added a high level sentence to point out that our data provide considerable new understanding of sub-surface particles (rather than those detected on the surface which is highly biased toward low density polystyrene), and that our infra red (FTIR) based elucidation of weathering processes provides a significant advance in environmental microplastics science.

L89-91: The results of this study fall however right in the range of the recently published study by Tekman et al. 2020 (doi: 10.1021/acs.est.9b06981), which should be mentioned in this context. Tekman et al. 2020 used 32 µm filters for sampling being therefore comparable to the 63 µm mesh used in this study. While Tekman et al. 2020 used the superior analysis technique of µFTIR imaging, they were unable to clearly differentiate between fibres and particles which is an advantage of this study.

Thank you, we included the study of Tekman et al 2020 in the discussion (L110-113)

L110-118: Information is missing on how many datapoints (n), being confirmed microplastics, fibres, PET fibres, were considered. The authors only state the number of samples considered but do not state here whether data were taken from individual particles/fibres and if so how many or if averaged etc. per sample. *Two new columns have been added to Table 1 so as to clarify the absolute number of particles counted (Suspected MPs and confirmed MPs).*

Furthermore, I short remark on the scale of the weathering index (e.g. the higher the value the more weathered the PET fibre?) would be preferable. It is also a bit irritating why this scale is opposed to the other two, namely having the high values in the violet colour section. *We have done as the reviewer suggested so as to provide three distinct datasets in the ODV Map (Figure 2) but in a more reader-friendly mode.*

L126: "of" missing: "This could reflect lesser inputs of MPs..." *Good catch! Done.*

L138: The following reference should be included: Ioakeimidis et al. 2016 (doi: 10.1038/srep23501). *Done.*

L147: Just one "I" for "Kruskal-Wallis". *Great catch! This reviewer read our paper ☐.*

L156: Maybe the relative size of the analysed subset (26.9%) could be stated here as well

We have now clearly detailed this in the methods and in our high level Methods summary paragraph early in the manuscript.

L168: It is not quite clear if the term “water-mass related” here is referring to a certain water body or the actual mass of particles/sample etc. I guess it refers to “water mass” and the hyphen should be erased or the term water body used. *Done.*

L168-177: Apart from the higher microplastic concentration in the surface water it is also interesting to point out that the polymer composition shows a larger variety compared to the deeper water masses. One more remark with regard to Figure 3 would be the composition of “paint”. Is it mainly polyurethane-based or acrylic-based paints? *Paint can be a complex mixture of both polyurethane-based or acrylic-based compounds, both categories have diverse spectra in our commercial FTIR library. We were not able to accurately identify the polymer source, but noted peaks that were characteristic of paint-like polymers.*

L219-220: In the study by Tekman et al. 2020 polyester was among the six most dominant polymer types.

This section discusses microplastic composition in the near surface water, though they note that the polyester polymers in Tekman et al. 2020 were particles. As synthetic polyesters are used exclusively in clothing, one explanation could be that polyester fragments in their study were products of fiber disintegration in the water column.

L219-223: The here mentioned publications by Bergman et al. 2019 and Peeken et al. 2018 also focus on small sized microplastics (down to 11 µm). Thus, the differences in the reported polymer types are more likely to be explained by the different analysis techniques (visual inspection + FTIR vs. µFTIR imaging) and the different reference databases used.

We thank the reviewer for this possible explanation, and have modified our interpretative language. Having mollified our language, we do feel it is important to relate our findings with those of others, with these two excellent studies providing a comparison of interest.

L248-254: The first three references have a different reference style than the others. This should be coherent. *Corrected.*

L292-294: same comment on incoherent reference style. *Corrected.*

L380: It is a bit unclear whether the total number of samples analysed was 71 or whether this is the number of analysed surface water samples and the 26 water column samples were additional samples. *Clarified.*

L391-401: In the light of the relatively low microplastic concentrations (102-103 particles m⁻³) it should be discussed how representative sample volumes of on average 0.07 m³ are.

Sampling microplastic particles in a mass of seawater entails a compromise between sufficient volume to detect and accurately report on concentration in a modest sample and collecting too much water that would lead to clogging of sieves or flow-through of smaller particles. In more remote locations, we strive for larger volumes (up to 460L), while in more urbanized (near source regions or in areas of high turbidity, we sample less (28L and up) or in depth related Niskin samples where volume is constrained (29 - 67L). Our substantial experience in sampling seawater allowed us a degree of comfort with these sample volumes, and our confidence in resulting identification and enumeration of microplastics was high. This confidence was supported by our adaptive correction of gross counts following exclusion of particles viewed as 'contaminants' of the process. We described this in the revised Methods that detail our approach.

L437-438: Does this mean that for all the particles sorted as “not microplastics” in this study, which were further analysed using FTIR spectroscopy, 10.7% were actually plastics and so on?

In our validation dataset, which includes all particles later flagged as contaminants, 10.7% of 'N' particles, 18.2% of 'U' particles, and 39.8% of 'Y' particles were plastic (revised methods, Line 537-547). Therefore, 39.8% of all suspect microplastics in our dataset were plastics, the remainder were non-synthetic (“N”) or particles for which our FTIR library returned no match nor where we were able to ascertain their identity via visual microscopy .

L440-445: Unfortunately, I cannot really follow the calculation of the microplastic concentrations since the variables used are not all defined clearly. Maybe it would be a bit more transparent if the data were provided as a table etc. It is also a bit misleading that concentrations labelled “FTIR-confirmed” seem to be estimations based on the actually confirmed microplastics + the number of suspected and not suspected microplastics that “should be” also microplastics based on identification rate? If this is the case, this needs to be made clear in the results/discussion!

We hope that our modified methods section clarifies reviewer's comment (Line 496-551). The power analysis was performed on a subset of particles drawn from the entire dataset (all samples across the Arctic and at depth) and validates the broad trends in microplastic composition across our region (Extended Data Table 1). It was not applied to the equation, which represents a method to arrive at microplastic counts per sample (MP m⁻³) corrected for false-positive and negative errors, and the rate of counting uncertain particles (those that cannot be identified as either plastic or natural).

For this calculation, we performed a complete FTIR analysis of suspect microplastics in 1 in every 4 samples, and additional randomly chosen particles from the remaining

samples. This amounted to 590 particles out of the 1570 that were used as a basis of identifying the error rates.

L488: What does “ample dataset” mean? What is the number of fibres/spectra used for this approach?

We changed the word ‘ample’ to ‘large’. We added two new columns to provide a sample size of enumerated and FTIR-analysed samples in Table 2. Polyester fibres dominated the composition in our samples – hence our opportunity to drill down and explore this dataset. Meaningful and worthwhile – in our opinion.

L491: An additional and fitting reference could be Tidjani 2000 ([doi.org/10.1016/S0141-3910\(00\)00039-2](https://doi.org/10.1016/S0141-3910(00)00039-2)).

We thank the reviewer for pointing out this interesting reference. However, we note that the study of Tidjani et al. 2000 is an in-depth characterization of the oxidation products generated from low density polyethylene exposed to UV irradiation, whereas our pilot assessment focuses on polyester fiber exposed to ambient seawater conditions. We feel its inclusion would detract from our Method and our interpretation without substantial elaboration.

L500: remove one “seawater” *Done.*

L516-518: different reference style. *Corrected.*

L562: “microplastics” should be abbreviated as “MPs” for coherence. *Corrected.* Does the term “sample” refer to the actual number of samples or particle/fibre? *Sample means sample – and this had been consistent through the manuscript. A sample is a sample of seawater ... MP particles were counted therein. We do not feel this warrants a change.*

REVIEWERS' COMMENTS

Reviewer #1 (Remarks to the Author):

Review of revision of "Arctic microplastics: polyester fibre profiles point to Atlantic origins" by Ross et al

In general, I am happy with the changes that the authors made to the manuscript. However, I still have three comments that I think the authors should address:

- Their new paragraph about the method in the main text is good, but does not state that only 27% of the samples were FTIR-analysed. It's disappointing that the authors have not been able to push this to 100% and remove any uncertainty about their results. At least, the authors need to be upfront about this in the main text
- They have not adequately responded to my original main comment 3. I strongly suggest that the authors redo their calculation with only the larger sizes plastic (i.e. cut off the size density function), so that they can confirm whether it's the smaller sizes that leads to the increase in total number.
- I still think that showing the ODV-interpolated maps in Figure 2 is over-interpreting the data, and that it would be much more scientifically robust to just show a colored scatter plot of the individual measurements. I leave it to the editor to decide whether the interpolation in Figure 2 is acceptable for the journal

Reviewer #2 (Remarks to the Author):

The authors addressed all raised questions and concerns on my behalf thoroughly and I am very pleased to suggest the manuscript for publication in Nature Communications. There are only two small comments I would suggest for revision (referring to the clean version of the manuscript):

L19-20: In my opinion, this statement has become less clear after rephrasing. I would suggest to go back to the original phrase.

L40, 43, 84, 108, 170, 244 and 266: the abbreviations "MP" and "MPs", respectively, should be used for consistency.

As a small note on the side regarding microFTIR imaging, several advances have been made recently to speed up the analysis. As recently highlighted by Pimpke et al. 2020 (<https://doi.org/10.1177/0003702820921465>) the time demand to scan a complete filter area of 14 mm x 14 mm is four hours when using a focal plane array (FPA) detector. The advantage of microFTIR imaging however, is the unbiased analysis of all particles on the scanned filter area independent of a visual pre-evaluation. However, I agree with the authors that microATR-FTIR provides usually a high resolution and good spectral quality making it a suitable alternative.

Reviewer #1 (Remarks to the Author):

Review of revision of “Arctic microplastics: polyester fibre profiles point to Atlantic origins” by Ross et al

In general, I am happy with the changes that the authors made to the manuscript. However, I still have three comments that I think the authors should address:

- Their new paragraph about the method in the main text is good, but does not state that only 27% of the samples were FTIR-analysed. It's disappointing that the authors have not been able to push this to 100% and remove any uncertainty about their results. At least, the authors need to be upfront about this in the main text.

We have detailed this in the Methods section (and updated from 27% to 37.6%). This is far above the minimum 10% used in most other studies. This is incredibly labour-intensive. Actually conducting FTIR analysis on 100% of our extensive samples would cost us an additional 1.5 years! Doing a subset may feel less informative but it is a routine formula used by virtually all groups today. As per the request of the Reviewer, we modified in the Results and Discussion to read:

“ Micro-FTIR analysis of suspected MPs (37.6% or 590 of 1570 of SMPs were analysed by FTIR) enabled an adaptive approach to estimating total abundance within samples and across the Arctic, while eliminating those particles identified as (non-plastic) contaminants.”

- They have not adequately responded to my original main comment 3. I strongly suggest that the authors redo their calculation with only the larger sizes plastic (i.e. cut off the size density function), so that they can confirm whether it's the smaller sizes that leads to the increase in total number.

I think the main point here is that it is difficult to compare across studies as each lab group has different methods and different minimum size of detection for microplastics. We entertain citations for other Arctic studies, but prefer at this point to not interpret too strongly. We softened our language on this point and moved this partial interpretation forward in the text from the conclusions paragraph to a more suitable spot on:

Lines 88-93:

“The size-frequency of MPs and SMPs revealed a skewed distribution which may partly reflect our choice of mesh size for sample collection (63µm; Supplementary Information Figure 2).

Our near-surface microplastic counts from sites across the Arctic fall into the range of those reported recently for Hudson Bay and the Eastern Arctic^{4,7,8}, but are higher than those measured

in the polar waters of south and southeast region of Svalbard, Norway⁶.”

Figure legend has been added to enable the reader to compare numbers on the basis of each size fraction as below:

“Supplementary Information Figure 2| Size frequency distributions for Suspected Microplastics (SMPs; left) and Microplastics (MPs) confirmed using Fourier Transform Infra Red Spectrometry (FTIR; right). Our use of a 63 μ mesh size at sampling may have increased retention of MPs slightly relative to other studies employing a 330 μ mesh size in the field, since 23% of SMPs and 19% of MPs fell below the 330 μ mesh.

- I still think that showing the ODV-interpolated maps in Figure 2 is over-interpreting the data, and that it would be much more scientifically robust to just show a colored scatter plot of the individual measurements. I leave it to the editor to decide whether the interpolation in Figure 2 is acceptable for the journal.

We have redone the colours in the ODV maps in Figure 2 in keeping with the request of the Editor. We feel that this is a good improvement over the previous version. We strongly prefer this version over that suggested by this reviewer as previously argued in our first rebuttal. Why? Because our current Figure 2 presents both univariate analyses (right hand XY plots with statistical support) plus the colour ODV maps which illustrate these data in a different manner. Call it a cartoon or an extrapolation. The version suggested by the reviewer is very weak in terms of visual appeal or understanding. As before, we are happy to defer to the better judgement of the Editor on this one, and present our preferred Figure 2, and the version suggested by Reviewer 1 (alternate).

Reviewer #2 (Remarks to the Author):

The authors addressed all raised questions and concerns on my behalf thoroughly and I am very pleased to suggest the manuscript for publication in Nature Communications.

There are only two small comments I would suggest for revision (referring to the clean version of the manuscript):

L19-20: In my opinion, this statement has become less clear after rephrasing. I would suggest to go back to the original phrase.

We have split this cumbersome sentence into two and hope this works better.

L40, 43, 84, 108, 170, 244 and 266: the abbreviations “MP” and “MPs”, respectively, should be used for consistency.

Thank you. Corrected as requested.

As a small note on the side regarding microFTIR imaging, several advances have been made recently to speed up the analysis. As recently highlighted by Primpke et al. 2020 (<https://doi.org/10.1177/0003702820921465>) the time demand to scan a complete filter area of 14 mm x 14 mm is four hours when using a focal plane array (FPA) detector. The advantage of

microFTIR imaging however, is the unbiased analysis of all particles on the scanned filter area independent of a visual pre-evaluation. However, I agree with the authors that microATR-FTIR provides usually a high resolution and good spectral quality making it a suitable alternative.

An excellent update and point. We have added this reference, and point to the 'opportunity' offered by technological advances, but also the limitations at present.

We have added the following:

“The use of focal-plane array detectors in FTIR analysis (FPA-FTIR) offer the opportunity to perform polymer source imaging on filter areas or whole filters without the need of pre-sorting of microplastics under the light microscope^{4,61}. While FPA-FTIR offers considerable potential to enhance throughput in microplastic analysis, it remains a recent development⁶².”